# Physical Activity of Polish and Norwegian Local Communities in the Context of Self-Government Authorities’ Projects

**DOI:** 10.3390/ijerph16101710

**Published:** 2019-05-15

**Authors:** Anna Maria Urbaniak-Brekke, Beata Pluta, Magdalena Krzykała, Marcin Andrzejewski

**Affiliations:** Department of Methodology of Recreation, Poznan University of Physical Education, 61871 Poznań, Poland; urbaniakanna@op.pl (A.M.U.-B.); krzykala@awf.poznan.pl (M.K.); andrzejewski@awf.poznan.pl (M.A.)

**Keywords:** physical activity, socioeconomic factors, local government, comparative study, action model

## Abstract

The aim of this paper is to evaluate the relationship between the physical activity (PA) undertaken by two groups of residents living in Poland and Norway, and projects run by their respective local authorities. A secondary goal is to determine PA levels in the studied social groups. Two representative groups (one from each country) were examined using a diagnostic survey, supported by questionnaires and interviews. The Polish cohort consisted of 382 respondents who were residents of 11 municipalities in the Kalisz district of the Greater Poland voivodship. The Norwegian cohort was made up of 378 residents of 8 Indre Sogn municipalities from the Sogn og Fjordane region. Norwegians are twice as physically active as Poles and assess their municipal sport and recreation facilities to be much better. There is no statistically significant relationship between Polish and Norwegian PA levels in the two studied groups and their positive views on the impact of their local governments’ projects to promote PA. Statistically significant correlations occurred between the frequency of PA undertaken, the time pattern of class unit, MET level (metabolic equivalent) and the declared use of the sport and recreation facilities of the two communities. Both groups are more willing to be active outdoors than indoors and thus municipal authorities should take particular care about the state of outdoor sports and recreation infrastructure. An innovative and original action model is presented to assist local authorities in their attempts to raise PA levels in their communities.

## 1. Introduction

Physical activity (PA) is very important due to the many benefits derived from maintaining its appropriate level. This is particularly the case for adults who, unlike children and adolescents participating in structured physical education (PE) classes, do not have such an explicit external motivator. Regular PA provides a number of positive health benefits and reduces the risk of many diseases in children and adolescents [1] and in adults [2,3]. PA guidelines vary, stipulating at least 60 minutes of moderate-to-vigorous physical activity per day for children and at least 150 minutes per week for adults [4].

Among many other contributing factors, local authority projects can affect the daily PA undertaken by their residents. PA levels in (non-urban) local communities are rarely the object of scientific interest or research, despite the fact that in many countries, including Poland and Norway, a significant portion of society lives outside large agglomerations and urban centres. As Biddle and Mutrie [5] rightly observe, PA-related behaviour is, in most cases, conducted in a societal context, decisions (individual or group) which are strengthened or weakened by the social environment, of which the actions of local government organizations and institutions are an integral part. 

In Poland, the responsibility for physical culture projects has been legally transferred from the state government level to the municipal one. Under new legislation in this area, it is municipal governments who have become the bodies obliged to run campaigns aimed at developing infrastructure and making their local communities more active [6]. Previous research conducted on this subject has critically assessed the activities of local governments in Poland. About 20% of Poles think that the possibilities of being active offered in their place of residence are insufficient, and that local authorities do not create the right settings and conditions for such activity [6]. Analysing research on the role of local administration in promoting PA in Norway, organisations including The Norwegian University of Science and Technology in Trondheim (Norges Teknisk-Naturvitenskapelige Universitet-NTNU) and the Nord-Trøndelag Health Research Study Centre (Helseundersøkelsen and Nord-Trøndelag-HUNT) have defined factors influencing the decision of PA take-up by residents of local communities. They have also defined local authority projects as those undertaken by the state government, local governments, public policies or by general concept of social support as those of the highest importance [7]. It has also been emphasized that it is important for any initiative to have input from all sectors: public, private and voluntary [7]. Another important scientific report, this time from Vestfold College, is a collective work from 2013 which describes the results of a 3-year project Active in the Open Air (Aktiv and friluft). Particular emphasis was placed on meeting the challenges related to the development of effective and sustainable cooperation between local authorities and non-government organizations (NGOs) in terms of PA. Evaluation of the project indicated that such cooperation had great potential, including promoting physical activity among people who are not yet active. The report also recommended that NGOs should cooperate with each other in order to strengthen their voice when communicating with local governments [8].

Scientists from other European countries have also carried out research on local authorities’ activities and their influence on PA level in their communities. Most works published in English come from Great Britain, where bodies such as *Sport+ Recreation Alliance* and *Sport England* create programmes aimed at activating local citizens irrespective of their social or financial status [9,10]. The authors of the programmes place a strong emphasis on their cooperation with local governments, which, they argue, play a key role in increasing PA levels among citizens. The Local Government Association in England, which also deals with sport and health issues, claims that encouraging adults to be physically active through sport, and encouraging children to play actively, has a positive influence not just on their health, but also their social interactions, ability to learn, safety, and economic situation. Finland is another European country where the role of local governments is taken into consideration. Extensive research has been carried out there in the form of the *Finnish Local Government: Sport Services* report, in which local government workers were asked about their scope of duties and responsibilities, as well as about the kind of services and sport facilities available in their municipalities and who was in charge of them [11].

The aim of this study is to determine the relationship between PA undertaken by a group of citizens from Poland and Norway and their respective local governments’ PA campaigns. A further aim is to define the PA levels of the surveyed social groups in the two countries and to propose an action model for municipality governments in terms of creating PA-related behaviour.

The reason for choosing Norway as a field of comparative research is the high level of physical activity of this country’s population and the ability to trace the processes led to this thanks to existing research. In Poland, there has been a steady increase for several decades in public interest in PA. The basic objective of the comparative research in both countries is to become acquainted with the processes, directions and development pace surrounding PA, both in the scope of awareness and in undertakings, which seems to be very important for full understanding and optimization of activities promoting the development of physical activity in local areas. However, one could ask whether the comparison between Poland and Norway, given that they present diverse levels of socio-economic development, makes sense. If a number of current social, environmental and economic indicators are analysed, however, both countries display a high level of social development. These include the level of medical services, quality of the environment, access to basic and higher education, organization of leisure time, human rights, security and tolerance in society, [12].

## 2. Materials and Methods

### 2.1. Participants

The study sample was selected randomly and participants came from various social classes. To achieve this, 2016 data [13,14] published in the Polish Central Statistical Office and Statistik Sentralbyra in Norway were used. 382 participants from Poland (citizens of 11 municipalities in the Kalisz district of the Greater Poland region) and 378 respondents from Norway (citizens of 8 municipalities in the Indre Sogn district of the Sogn og Fjordane region) participated in a survey. The compared groups did not differ in terms of sex, age and professional status. Among the Polish respondents 37.8% held secondary education certificates, while only 33.2% of Norwegian respondents held a BA degree, a statistically significant difference. Most Polish respondents claimed to have a moderate social status (51.7%), while most Norwegians claimed to have a good one (48.9%). The duration of residence in the municipalities also turned out to be statistically significant. Most Polish respondents had lived in their area for a minimum of 6 years (94.5%), while in Norway the percentage was lower (85.6%).

All procedures were in accordance with ethical standards. The study was conducted in compliance with the Declaration of Helsinki and was approved by the local ethics committee (No. 1095/15. The study protocol was also approved by the Local Board of Ethics of Karol Marcinkowski University of Medical Science, in Poznań. Participation in the study was voluntary, and participants were informed that they were free to refuse discussion of particular topics or to end their interview at any time. Confidentiality was maintained by using pseudonyms and coded personal data information.

### 2.2. Procedures

The research process was based on a diagnostic survey supported by questionnaire and interview techniques, as well as on the analysis of local authority documents. A questionnaire by Basińska-Zych [6] was used. The questions were divided into 3 parts. The first related to the awareness of PA role and its importance, the second to the forms and frequency of PA undertaken, and finally to the assessment of local government projects aimed at increasing PA levels among its residents.

Interviews with representatives of all 19 of the surveyed municipalities in Poland and Norway were open, in-depth and consisted of questions related to the following: evaluating municipality authorities’ attractiveness in terms of its PA policy, assessing local government projects that support citizens in taking up PA, and cooperation with other local governments, non-government organizations and citizens, as well as evaluating tools applied by local governments to check whether their activities and policies actually increased PA levels amongst its citizens.

To validate the research tools, a translation procedure was used and the psychometric properties of the translated text were used. A further indispensable element of validation was cultural adaptation, which allowed for intercultural comparisons and practical application of questionnaires in both countries. The equivalence of the adapted tool with the original version was measured through five equivalence criteria: facade, psychometric, function, translation and accuracy reconstruction [15]. To measure internal reliability, Cronbach’s coefficient-Alpha was calculated for the question scales; the reliability of the total was 0.87. The questionnaire has acceptable psychometric properties including evidence of construct validity, internal consistency (α = 0.85) and test-retest reliability (r = 0.89). These values indicate good internal consistency for all the question scales.

To operationalize the tested variables, the following indicators were used: MVPA (Moderate to Vigorous Physical Activity) requiring moderate effort and causing cardiac acceleration (3–6 MET-Metabolic Equivalent) and VPA (Vigorous Physical Activity)—physical activity of high intensity, requiring considerable effort, causing faster breathing and a significant increase in heart rate (>6 MET). The level of MVPA and VPA was determined with a Physical Activity Screening Measure [16].

### 2.3. Statistical Analyses

The significance of differences between two variables, of rank and quantity, were checked using the U Mann-Whitney. The significance of differences between more than two mean values was checked using the Kruskal-Wallis test. The significance of the differences between the qualitative (nominal) variables was checked using the chi square independence test. Correlations between variables were verified using the Spearman rank correlation coefficient. In statistical analyses, the significance level *p* = 0.05 was assumed. The analyses were performed using the SPSS programme.

## 3. Results

The study sample characteristics with regard to the variables included in the analysis are summarized in Table 1.

### 3.1. Physical Activity Level of the Research Group

The PA level of each respondent (MET-min/week) was calculated based on the frequency of PA undertaken, duration of exercise, and preferred forms. A standardized procedure was used, as described in the work of Biernat et al. [17] and Ainsworth et al. [18]. The values of the MET coefficient were assigned assuming that <3 MET means low activity, 3–6 MET moderate-to-vigorous physical activity (MVPA), and >6 MET vigorous physical activity (VPA). Furthermore, measures of central tendency and dispersion for the product of the PA level factor were determined (Table 2).

The difference between country samples is statistically significant. The mean values of PA units expressed in MET-min/week indicate a significantly higher PA level in the Norwegian cohort. Based on current health recommendations [18], the Norwegian respondents fall into the high activity group (>3000 MET-min/week). The standard deviation from the average value is much larger among Polish respondents than among Norwegians. The median values in both cases are distant from the values of the calculated average, which proves the diversity of the provided responses. Other measures also indicate a high heterogeneity of responses. The most common forms of PA are walking in the neighbourhood (63.2%) and cycling (45.3%). In no case did the percentage of people declaring any other forms of PA exceed 24%. Poles more often preferred cycling and swimming. Norwegians, however, walked in the woods, went to the gym, did aerobics, ran, kayaked and went skiing or ice-skating more often than Poles. In both groups, the analysis showed statistically significant relationships between independent variables and the frequency of PA exercised (Table 3).

In the Polish group, younger people and people with a good financial situation were more physically active. In the case of the Norwegian group, younger people and people with higher education levels were more likely to undertake PA.

### 3.2. Local Self-Government Actions in Terms of PA

According to the respondents, local governments frequently organised sport and recreation events (48.3%) and co-financed recreation activities for children and adolescents (43.4%). They rarely subsidized leisure time activities for adults (19.5%) or created education campaigns for a healthy lifestyle (13.4%). Differences in opinions of respondents from Poland and Norway proved to be statistically significant (Table 4).

In both countries, a proposed list of sport and recreation activities was highlighted as an important local authority activity supporting the development of PA. Only 26.9% of the total number of respondents participated in sport and recreation activities financed or subsidised by local governments. The relationship between the distribution of responses and the country of residence was quite significant (χ^2^ = 54.64. *p* < 0.001). The respondents from Norway (39%) more often used these types of activities than those from Poland (15%).

The economic situation, i.e. public expenditure (Figure 1), had a considerable impact on the PA activities undertaken by local authorities.

The results of the above study are particularly diverse and, interestingly, these differences do not appear clearly between the countries surveyed, but rather between individual local governments. Financial and investment policy issues in local governments were also analysed from the residents’ perspective by examining their opinion on local authority subsidies to sport and recreation NGOs. Nearly 57% of all respondents thought that this sort of funding existed and was successfully realized by local governments. The difference for both countries turned out to be statistically significant (χ2 = 9.10, *p* = 0.003). On the basis of the conducted research, the distribution of responses was not dependent on gender, age, education, occupational situation, financial situation or residence duration in the municipalities.

### 3.3. The Action Model of Local Self-Governments in Terms of Creating PA-Related Behaviour

Based on the analysis of the research results and information collected in two local environments, a local authority action model was created and proposed, aimed at stimulating the growth of PA among local populations (Table 5).

The action model is of an implementational nature and consists of fourteen steps, syntactically arranged, which allows for the change in the order, depending on the current situation and the possibilities of the particular self-government.

## 4. Discussion

### 4.1. Level of the Physical Activity in the Studied Groups

The main aim of the research was to assess the PA levels of the studied cohorts in Norway and Poland in the context of activities undertaken by their local authorities. The weekly average PA level (expressed in MET units) for Polish cohort was half that (1739 MET) of those in Norway (3378.9 MET).

Eurobarometer data from 2013, in which PA levels of inhabitants in various European Union countries was analysed, indicates that younger men, with higher education and good financial standing from rural areas of northern Europe obtained the highest MET level [18]. This finding is directly confirmed by the results presented herein. Norway and Poland can both be categorised as highly developed countries. This is evidenced by, among other things, the Human Development Index (HDI), calculated by UNDP (United Nations Development Program), which determines not only the level of economic development expressed as real income, but also takes into account non-economic factors such as life expectancy and access to education and culture (including physical activity). Out of the 182 countries on the HDI, both Norway and Poland are among the best developed in the world [19].

PA frequency and duration in the described groups differs greatly. Statistically, respondents from the Indre Sogn district declared that they undertook PA much more often than Polish respondents and that this activity usually lasted longer. Similar conclusions have been drawn by the surveys of the Central Statistical Office in Poland, presented in its *Participation in sport and physical recreation in 2016* report. It was shown that the most physically active group were young people (up to 14 years 82%, and 15–19 years 71.2%) with lower secondary and higher education (69% and 62% respectively) [19]. The obtained results are also confirmed by data published by the Ministry of Sport and Tourism, as well by Kantar Public as part of the 2017 report on *The Physical Activity Level of Poles*. The International Physical Activity Questionnaire (IPAQ,) was used to check whether Poles meet the PA standards recommended by the World Health Organization (WHO). It has been proven that the “percentage of people meeting the WHO recommendations in each individual age group decreases with age—the older the age group, the smaller the proportion of people who meet the recommendations of the World Health Organization in their leisure time or during the activity related to the journey between home and work, school or shops” [20]. Here, it is also worth mentioning that according to a study published in 2018 on promoting physical activity, less than 50% of EU Member States have “developed policies for the ‘Senior Citizens’ sector” [21]. It was found that a greater number of men (18.9%) than women (13.4%) take part in PA at the level suggested by the WHO [20], which also corresponds to the research results presented in this paper. At the same time, according to an earlier report from the same company (Public Kantar), in 2016 approximately 21% of Polish men and 16% of women were physically active [22].

The results also show that the number of surveyed Norwegians participating in PA corresponds almost exactly with the data presented in the work of Breivik and Hellevik [23], who studied the changes in PA level of Norwegians between 1985 and 2011. They found that PA levels in the Norwegian population increased within the studied time frame, yet still only 4 out of 10 people (about 40%) took part in PA three times a week, and only 1 in 5 of those surveyed did so five days a week [23].

In 2012, *The Lancet* magazine published the *Global Physical Activity Levels* report containing PA research results for people around the world. They showed that 31.3% of the global adult population does not systematically undertake any PA. The best results were found in inhabitants of Southeast Asia (only 17% of people are inactive), and the worst in both North and South America and the eastern Mediterranean (where 43% of people do not undertake regular PA). PA levels decrease with age and are usually lower for women than for men [24]. The authors of *Global physical activity levels* claimed that the level of overall PA is lower in high-income societies (e.g., North America), while in the research for the present paper it has been proven that PA levels are twice as high in Norway, whose inhabitants are characterized as undoubtedly having a better economic situation than Poles. Similar conclusions are drawn by Veal, who [25] refers to the theory of the Veblen effect (an increase in the demand for luxury goods, despite an increase in prices) and suggests that the more affluent the society, the higher the PA level and greater the interest in this subject matter [25]. Detailed data presented in *Global physical activity levels* indicates an increase in PA levels during leisure time in recent years, mainly in terms of participation in sports (these results have subsequently been confirmed through studies carried out in Canada, Spain, Sweden and England, countries with a relatively high standard of living). The results of studies carried out under *the Healthy People* programme also correspond to the above findings. In the USA, the percentage of adults regularly taking moderate LTPA up to 150 minutes per week, or vigorous up to 75 minutes per week, increased from 43.5% in 2008 to 49.9% in 2014. Additionally, the percentage of people regularly undertaking moderate activity up to 300 minutes per week or vigorous activity up to 150 minutes per week increased from 28.4% in 2008 to 34% in 2014 [2,26].

The research conducted for this paper has shown that there is no statistically significant relationship between the PA levels of residents (expressed in MET) in Poland and Norway and their positive view on the impact of local government activities to promote PA. However, statistically significant dependencies occurred between the PA frequency, duration of a single class unit, level of MET and the declared use of the sport and recreation facilities. People who trained using the equipment and facilities available in their local area were more active for a longer period and at a higher MET level. This means that local governments can indirectly shape the PA of their citizens through appropriate investment in developing and improving sports and recreation infrastructure.

### 4.2. Local Government Policy in the Field of the Physical Activity

Based on the information collected from interviews with representatives of local authorities and an analysis of documents, it can be stated that the budgets of all the analysed local authorities in Poland and Norway include activities aimed at promoting and developing PA among their residents, although amounts vary significantly between individual authorities. The average percentage for the surveyed Polish local governments is 1.20%, while for Norwegian ones it is 1.09%. This means that in Polish municipalities a slightly larger amount of the budget was allocated to PA than in Norwegian units.

The same applies to supporting organizations and institutions operating in the sphere of PA. The main area of investment and financial aid from the municipality office, and extra-budgetary resources gained through the efforts of this office, is in the expansion and maintenance of sports and recreation infrastructure, organization of sports events, providing financial support for local sports clubs and also sponsoring awards for competition winners [27,28]. The development of other elements, such as the appointment of trainers and instructors of recreation activities or the promotion of a healthy lifestyle, does not have the same high priority. The research results were additionally supplemented by data published in the extensive report *Physical Activity: The scope, facilities and social inequalities*, which analyses, among others, the amount and purpose of financial resources allocated from the budget of the Kingdom of Norway to PA. In 2010, twice as many financial resources (*spelemidlar*-spel funding) were allocated for purposes related to sports and recreation infrastructure than in 1985. In the period between 2000 and 2011, more than 70% of the allocated resources were used for building football facilities, multi-purpose halls and swimming pools. At the same time, the authors of the report emphasize that many tasks were not carried out due to insufficient resources (e.g., road shoulders/pavements or bicycle paths which, because of rocky terrain, require complicated construction work). They conclude that it would take at least 20 years for a network of hiking and cycling paths in Norway to reach a satisfactory level [29]. At the same time, it is worth mentioning that priority is given in Norway to investments related to the development of sports and recreation infrastructure of an organized and professional nature (sports fields, sport halls). Infrastructure and facilities designed to enhance PA in an unorganized way (tourist trails, cycle paths) receive comparatively little support [29].

It is worth emphasizing that in Polish municipalities the PA development strategy and the budget report are two separate documents. By contrast, in Norway, both planning and financing are included in plans for PA (Kommunal plan for fysisk aktivitet). It is important to stress that in almost every district (except for cases of very financial dire straits) part of the budget is allocated to activities related to residents’ PA, which in turn is likely to positively influence the rise in PA level.

The local government action model presented in this work is a useful tool for municipal administration employees for raising PA levels among residents. The most important recommendation for local authorities (and indeed for other organizations and institutions operating in the PA field) is the maintenance of sports and recreation infrastructure and of green and recreation/leisure areas. These elements of an authority’s scope of actions have proven to be of key importance in allowing residents to spend their free time actively. This is why it is primarily recommended that authorities invest in outdoor infrastructure (including cycle paths, marked tourist routes or outdoor gyms), because, as shown by research results, forms of PA undertaken in the open air enjoy a much greater level of take-up by residents in the areas studied than those performed indoors. In addition, it is recommended that training sessions and workshops for representatives of municipal offices are organised, informing them about the possibilities and obligations of local governments in terms of their residents’ PA. This model will be especially helpful in the work of departments dealing with public health, sport, investments, local development, and EU projects. It indicates solutions that will have a positive impact on the PA level among residents in a given municipality. The recipients of the research results included in this work are primarily employees of rural and urban-rural local governments. The results are addressed primarily to those responsible for actions related to public health, health promotion and sports and recreation activities. The guidelines included in the presented model can also be used by people working in other public sector units, as well as by representatives of private enterprises and NGOs undertaking cross-sectoral cooperation on increasing societal PA levels.

### 4.3. Limitations

In summarising the research carried out here, there are also some limitations that need to be indicated. First of all, it should be remembered that the research was of a cross-sectoral nature, which does not allow for definite statements about causal relations between variables. However, having the belief about the high cognitive value of the obtained results, the highly probable character of the described relations should be highlighted. Another limitation of the research was the use of declarative PA measures based on self-reporting. Although this is an approach commonly used in population studies, it carries the risk of some bias.

## 5. Conclusions

The work concerns international comparative research and pursues two basic goals: practical-social and theoretical, including the reporting on the differences of the studied phenomena associated with PA levels of residents of local communities. The originality of the research is manifested not only in it highlighting the important role of the authorities in the promotion and support of sports and recreational activities, but also in examining the issue from an international perspective; that is, in two separate cultural, legal, organizational and linguistic contexts.

Nowadays, attempts are being made to implement international comparisons in various areas of social life. This also applies to comparisons between European countries with a differentiated level of economic development. The primary purpose of such research is to learn something more about ourselves through a better understanding of others. They constitute the main area of research in comparative sociology which deals with the systematic and orderly compilation and comparison of phenomena and processes occurring within different societies. The authors of this paper have investigated an extremely important problem: the assessment and promotion of PA of people living in two countries. Similar international comparative research should also be carried out in relation to other spheres of social life.

The activities of local authorities are among many factors affecting daily physical activity. This influence is particularly important in the smallest units of local government because they are the units closest to their inhabitants and thus know most about the conditions and trajectories of local development. The situation in urban and urban-rural districts, far away from large agglomerations, is worth exploring, both from a residents’ perspective and that of representatives of local authorities. There is a great number of Polish and Norwegian citizens living in such districts but there is little research on them. In order to increase the value and reliability of the results, analysis was carried out simultaneously in Poland and Norway. In this way, it is also possible for Poland to draw inspiration from another country, which is at a high level of economic and social development and which offers interesting solutions in the field of PA. Based on the research, an original action model for commune authorities was presented, aimed at increasing the PA levels in given local districts. The results of the research are important from the point of view of public health and indicate the directions in which health promotion policy should follow, including promotion of the physical activity of the local community. In order to optimize the strategy of promoting physical activity in local communities and adapting them to environmental conditions in Poland and Norway, it is necessary to increase financial support for organizations and non-governmental institutions operating in the sphere of physical culture. The recipients of the presented research results are, first of all, public administration employees responsible for taking action in the field of public health, health promotion and sports and recreation.

## Figures and Tables

**Figure 1 ijerph-16-01710-f001:**
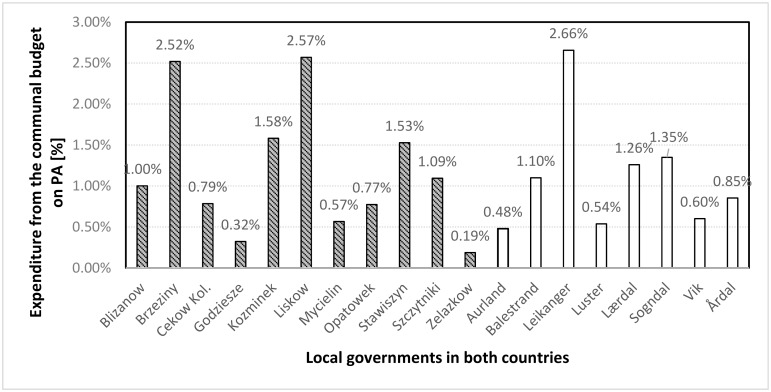
Funds allocated to PA-related activities in general expenditure from the budgets of the local governments in Kalisz and Indre Sogn districts (2010–2015).

**Table 1 ijerph-16-01710-t001:** Research group analysis.

Variables		Poland		Norway		Chi Square Independence Test Independence Test Square
		*n*	*n* %	*n*	*n* %	*p*
Sex	women	191	50.0	186	49.2	0.827
	men	191	50.0	192	50.8	
Age (years)	15–19	32	8.4	31	8.2	0.156
	20–34	104	27.2	83	22.0	
	35–49	96	25.1	87	23.0	
	50–64	86	22.5	89	23.5	
	65 and more	64	16.8	88	23.3	
Education	primary	27	7.3	13	3.5	<0.001 *
	vocational	64	17.3	64	17.3	
	secondary	140	37.8	96	25.9	
	with Bachelor’s degree	56	15.1	123	33.2	
	with Master’s degree	83	22.4	74	20.0	
Work experience	student	40	10.7	44	11.8	0.340
	worker	249	66.8	261	70.0	
	unemployed	84	22.5	68	18.2	
Average income per person	less than PLN500	29	7.9	17	4.6	<0.001 *
	PLN500-1000	150	41.1	59	15.9	
	more than PLN1000	186	51.0	295	79.5	
Financial situation	very good	16	4.3	80	21.3	<0.001 *
	good	137	36.5	184	48.9	
	moderate	194	51.7	88	23.4	
	bad	25	6.7	19	5.1	
	very bad	3	0.8	5	1.3	
Residence (years)	less than 2	3	0.8	24	6.4	<0.001 *
	2–5	18	4.8	30	8.0	
	6–10	52	13.8	30	8.0	
	more than 10	305	80.7	291	77.6	
Municipality	Blizanów	27	3.6	23	3.0	Aurland
	Brzeziny	46	6.1	18	2.4	Balestrand
	Ceków Kol.	23	3.0	30	3.9	Leikanger
	Godziesze Wlk.	42	5.5	67	8.8	Luster
	Koźminek	34	4.5	29	3.8	Lærdal
	Lisków	23	3.0	100	13.2	Sogndal
	Mycielin	23	3.0	36	4.7	Vik
	Opatówek	50	6.6	75	9.9	Årdal
	Stawiszyn	34	4.5			
	Szczytniki	38	5.0			
	Żelazków	42	5.5			

* statistically significant differences.

**Table 2 ijerph-16-01710-t002:** Measures of central tendency and dispersion for the MET coefficient for all respondents.

Country	*M*	*SD*	*Me*	*Max*
Total	2554.7	3205.74	1440.0	22560.0
Poland	1739.0	2556.35	780.0	17640.0
Norway	3378.9	3567.95	2160.0	22560.0
Mann-Whitney U test	*Z* = −8.79; *p* < 0.001 *			

*M*-mean, *SD*-standard deviation, *Me*-median, *Max*-maximum, *Z*-value of Mann-Whitney U test, *p*-level of significance, * statistically significant difference.

**Table 3 ijerph-16-01710-t003:** Situation variables x PA incidence.

Country	Age		Education		Average Income		Financial Situation		Residence	
	*CC*	*S*	*CC*	*S*	*CC*	*S*	*CC*	*S*	*CC*	*S*
Poland	−0.27	<0.001 *	0.08	0.139	0.10	0.052	−0.21	<0.001 *	−0.01	0.779
Norway	−0.10	0.046 *	0.10	0.047 *	0.08	0.113	−0.07	0.193	−0.06	0.219

CC-correlation coefficient, S-significance (bilateral).

**Table 4 ijerph-16-01710-t004:** Local government activities for PA development x country of residence (*n* %).

Actions	Country		Chi Square Independence Test	
	Poland	Norway	*χ^2^*	*P*
Organize sport and recreation events for residents	52.6	41.3	9.82	0.002
Arrange education campaigns for a healthy lifestyle	11.3	14.8	2.12	0.145
Invest in the development of sport and recreation infrastructure	35.3	37.3	0.32	0.574
Subsidise recreation activities for children and adolescents	29.6	55.0	50.42	<0.001 *
Co-finance leisure time activities for adults	9.4	28.6	45.36	<0.001 *
Maintain green and recreation areas	30.9	46.0	18.41	<0.001 *
Subsidise sport clubs, physical culture institutions	25.1	54.0	66.13	<0.001 *
Support the actions of sport and recreation centres. community centres	23.6	54.2	75.27	<0.001 *
No action initiated	8.7	2.1	15.91	<0.001 *

* statistically significant differences.

**Table 5 ijerph-16-01710-t005:** Recommended model of operation of local authorities for the purpose of raising PA levels in the population.

Action	Example	Citation
1. Appoint a physical activity coordinator within the municipal office	Public health coordinator in Norwegian municipalities	*Public Health Act, [Lov om folkehelsearbeid (folkehelseloven)]*, LOV-2011-06-24-29, §4, Ministry of Health and Care Services of Norway.
2. Create a planning group	The group in the Luster district (Norway) includes representatives of: public health coordinator, schools (especially from schools with sport facilities open to the public), the local tourist organisation, health services, social care institutions, the police, and local government departments for: finance, investment and development, culture, education, promotion, planning, projects, and extra-budgetary funds	*Municipal plan for physical activity and public health, [Kommunedelplan for fysisk aktivitet og folkehelse 2016-2019]*, Aurland district, www.aurland.kommune.no; Also suggested during an interview with public health coordinator in the Luster district.
3. Prepare a report on the current situation	Report including a presentation of the level of knowledge on the physical activity of residents, the condition and development of sports and recreation infrastructure, current activities and investments	*Association Between Sport Participation, Body Composition, Physical Fitness, and Social Correlates Among Adolescents: The PAHL Study*, International Journal of Environmental Research and Public Health, 2018; *The role of local governments in creating supportive environments for physical activity*, doctoral dissertation, James Cook University, Townsville, 2011
4. Create the action plan	Projects for the development of free-time PA of the local community (Norway)	*Strategy for Sports Development in Poland until 2015, [Strategia Rozwoju Gminy Ceków Kolonia na lata 2017-2021]*, Ministry of National Education and Sport, Poland; *Sport for All in the City’s Space*, Ministry of National Education and Sport, Poland, 2016; *Municipal plan for physical activity, sport and outdoor activities 2016-2027, [Kommunal plan for fysisk aktivitet, idrett og friluftsliv 2016-2027]*, Luster district; *Municipal plan for sport and physical activity 2016-2019, [Kommunal plan for idrett og fysisk aktivitet 2016-2019]*, Balestrand district
5. Arrange public consultations	Annual survey with residents, open meetings before the implementation of the local government’s plan for physical activity	Possibility of having input into the revision of the plan in the Luster district: www.luster.kommune.no/kommunedelplan-for-fysisk-aktivitet-idrett-og-friluftsliv-2016-2027-kunngjering-om-oppstart-av-planarbeid.6189107-157426.html; Possibility of having input into the revision of the plan in the Balestrand district: https://www.balestrand.kommune.no/kommunal-plan-for-fysisk-aktivitet-idrett-og-friluftsliv-hoeyring.6149493-165229.html; Also suggested during interviews with local government representatives in Szczytniki and Cekow districtss in Poland
6. Publish and implement the action plan	Plans and project descriptions are available on the municipal website and at their offices	*Municipality plan for physical activity, sport and friluftsliv 2016-2027, [Kommunal plan for fysisk aktivitet, idrett og friluftsliv 2016-2027]*, Luster district, available on the website: www.luster.kommune.no; *Municipal plan for sport and physical activity 2016-2019, [Kommunal plan for idrett og fysisk aktivitet 2016–2019]*, Balestrand district, www.balestrand.kommune.no
7. Control the plan	Committee for annual scrutiny of the plan (specialists of public health, physical activity and economy) or control by members of the planning committee	Suggested during an interview with the public health coordinator in the Luster district
8. Apply for extra-budgetary funds	Obtaining extra-budgetary funds (e.g., Marshal’s Offices, Ministries, EU-programmes), searching for new investors and financial supporters (e.g., through Local Action Groups (LAGs) and sponsorship opportunities)	NIVEA action ‘Family playground’ (Brzeziny and Szczytniki districtss); EU-projects (Program Rozwoju Obszarów Wiejskich), Opatówek district: https://www.opatowek.pl/cat,60,309, https://www.opatowek.pl/cat,60,320; ORLIK 2012 - project co-financed by Marshals’ Offices in Poland, the Polish Ministry of Sport and Tourism, and local governments (all Polish local authorities described in the article have ‘ORLIK’ facilities)
9. Cooperate with other municipalities	Work of Healthy Lifestyle Centres in Norway (Frisklivssentralen) in Leikanger, Luster and Sogndal districtss (Norway), LAGs in Poland	All the eleven Polish districts are gathered in two Local Action Groups (LAG): ‘The land of Nights and Days’ and ‘Osmunda Regalis’
10. Cooperate with specialists	Open lectures of specialists in the field of physical activity, health promotion, lifestyle, new forms of sport and tourism. Information about sport and recreation on the municipality website	Universities of Third Age in Poland; Information about culture, sport and free-time available on the official website of the Aurland district: https://www.aurland.kommune.no/; Information about sport and culture available on the official website of the Stawiszyn district: http://stawiszyn.pl/strona/
11. Cooperate with non-government organizations	Cooperation with non-governmental organizations and institutions, including: sports clubs, student sports clubs, tourist associations, foundations and other groups working for healthy, active lifestyles	*Best Practices in Sport Objects Managing,* Ministry of Sport and Tourism, Poland, 2017; *Basics of Organization and Management of Sports Institutions*, Polish Corporation of Sport Managers, Poland, 2000.
12. Invest in infrastructure	Modernization of sports and recreation facilities, being open to innovative solutions, consultations with specialists and residents	*Reports on the implementation of the budget of the Opatówek District,*http://www.bip.opatowek.pl; *Municipal plan for physical activity, sport and friluftsliv 2016-2020, [Kommunal plan for fysisk aktivitet, idrett og friluftsliv 2016-2020]*, Vik kommune, http://www.vik.kommune.no; *The role of local municipalities in development of tourist and recreational areas*, Katowice, 2013.
13. Arrange for classes and events	Organization of events, fairs and competitions, supporting school competitions, free admission to sports events (Polish and Norwegian municipalities)	*Municipal plan for sport and physical activity 2012-2024, [Kommunal plan for idrett og fysisk aktivitet 2012–2024]*, Leikanger kommune, http://www.leikanger.kommune.no.
14. Actively promote	Local and regional authority participation in events to promote PA (Polish and Norwegian municipalities)	*Physical activity; scope, facilitation and social inequality, [Fysisk aktivitet; omfang, tilrettelegging og sosial ulikhet]*, HIF-Rapport 10, Høgskolen i Finnmark, 2010.

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
