# Peer review of "Physical Activity of Polish and Norwegian Local Communities in the Context of Self-Government Authorities’ Projects"

_ijerph, 2019, doi:10.3390/ijerph16101710_

Round 1

Reviewer 1 Report

Dear Authors, 

I have no more comments on the new version. 

Regards

Author Response

Dear Reviewer,

Authors of the manuscript thank the Reviewer for taking the time to make a thorough revision to our manuscript and provide constructive comments. We feel that the review has helped to enhance the quality of the article.

Reviewer 2 Report

The conclusions need to be revised to focus especial to highlight the main ideas of your study.

The discussion need to be extending to underline the relevance of your findings related to previous researches.

Author Response

Dear Reviewer,

In response to Your comments in the review of the manuscript “Physical Activity of Polish and Norwegian Local Communities in the Context of Self-Government Authorities’ Projects”, I would like to inform you about the undertaken corrections and completions.

The conclusions need to be revised to focus especial to highlight the main ideas of your study.

The discussion need to be extending to underline the relevance of your findings related to previous researches.

The text has been revised following the Reviewer’s suggestion. The presented studies are innovative and do not have sufficient support in the literature. Especially when physical activity in rural areas of Poland and Norway is concerned. The authors have taken up an extremely important problem, which is the assessment and promotion of physical activity of people living in two countries. Similar international comparative research should also be carried out in relation to other spheres of social life.

As suggested by the reviewer, work style and grammar have been twice checked by a native speaker, and appropriate corrections have been applied to the manuscript language.

Authors of the manuscript thank the Reviewer for taking the time to make a thorough revision to our manuscript and provide constructive comments. We feel that the review has helped to enhance the quality of the article.

This manuscript is a resubmission of an earlier submission. The following is a list of the peer review reports and author responses from that submission.

Round 1

Reviewer 1 Report

Dear Authors,

Thank you for the paper titled '' Physical Activity of Polish and Norwegian Local Communities in the Context of Self-Government Authorities’ Actions''

The authors used the questionnaire of BasiĹ„ska Zych. 

This questionnaire was for the role of local government in tourism development in terms of the economic crisis – on the example of Wielkopolska Voivodeship.

So I think the Authors have adapted it to use in this paper. But There is no anything refers to the validity and reliability of the tool. ( I know that you made the psychometric properties of the translated text is also important).

In line 128 The significance of differences between more than two mean values was checked using Kruskal-Wallis test. I cannot find the table of these results. 

Also why the Authors did not calculate the effect size? 

Table 1 was presented in two pages, some data not clearly presented. 

Be sure to format the in lines 235-238.

In line 161 ''In the Polish group people at a younger age and with a good financial situation were more

162 physically active'' 

In the discussion section, there is no explanation of why?

Tn figure 1 you have to explain the axis title (X,Y).

Line 187 I would move these sentences to the discussion section ''The average percentage for the 186 surveyed Polish municipalities is 1.20%, while for Norwegian 1.09%. This means that in the Polish municipalities a much bigger amount of the budget was allocated to PA than in Norwegian units''.

From line  202 to 2013 most of them related to the discussion part, please reform it again if it is possible. 

Author Response

Dear Reviewer,

In response to your comments in the review of the manuscript “Physical Activity of Polish and Norwegian Local Communities in the Context of Self-Government Authorities’ Actions”, we would like to inform you about the undertaken corrections and completions.

Reviewer 2 Report

Dear authors, first of all, I would like to tell you that I think this is a very interesting piece of work. I would therefore like to make a number of suggestions for improving its scientific character:

Abstract

In this section it would be interesting to include the instruments they have used to collect the data.

Introduction

The introduction is very brief, you should expand it as there is currently a lot of information on this subject. I would also like to point out that the quotations used in this section are very old, and it is necessary for them to look for more current quotations, which provide the study with a current justification.

Method

This paragraph is very well expressed. However, it is necessary for them to include a section on instruments, in which they explain exactly what they have used to collect the data. Without this section, the scientific validity of the study is diminished.

Results

The results are very detailed. However, you must remove the color from the tables, and I suggest that instead of putting the numbers in red you put an asterisk.

In the same way, the tables must have a scientific format, that is to say, they must eliminate the vertical lines.

Discussion

The discussion is adequate. But as in the introduction, I encourage you to look for more current quotes with which to contrast the data you have found in the study.

Author Response

(The authors gave the same response as above.)

Reviewer 3 Report

However, the research gap based on existing body of literature is not clearly presented. What it was the novelty of study?

  Since the discussion is not opened to the reader, it is difficult to see how your study moves this field forward and what is the contribution you aim to make to the theory. This can also be seen in the discussion section where the theoretical and practical implications are omitted.

 You need to know the discussion and position your paper accordingly against this existing body of knowledge.

 The theoretical section should be able to explain the topic as well, to understand the phenomenon under study.

The results are not clear and convincing.  

The discussion section does not help to identify what the results provided in the light of existing body of knowledge that could bring the discussion forward. .

  How could the findings be theoretically generalized and therefore also applied to other contexts?

Author Response

(The authors gave the same response as above.)

Round 2

Reviewer 3 Report

The authors improved the article according with recommendations.